# Advanced Adaptive Cruise Control Based on Operation Characteristic Estimation and Trajectory Prediction †

**Hanwool Woo** [1,2,*] , **Hirokazu Madokoro** [1] , **Kazuhito Sato** [1] , **Yusuke Tamura** [2] , **Atsushi Yamashita** [3] **and Hajime Asama** [3]

[1]  Department of Intelligent Mechatronics, Faculty of Systems Science and Technology, Akita Prefectural University, Akita 015-0055, Japan; madokoro@akita-pu.ac.jp (H.M.); ksato@akita-pu.ac.jp (K.S.)

[2]  Institute of Engineering Innovation, Graduate School of Engineering, The University of Tokyo, Tokyo 113-0023, Japan; tamura@robot.t.u-tokyo.ac.jp

[3]  Department of Precision Engineering, Graduate School of Engineering, The University of Tokyo, Tokyo 113-0023, Japan; yamashita@robot.t.u-tokyo.ac.jp (A.Y.); asama@robot.t.u-tokyo.ac.jp (H.A.)

*  Correspondence: woo@akita-pu.ac.jp; Tel.: +81-1-8427-2000

†  This paper is an extended version of the paper published in the 2018 IEEE International Conference on Intelligent Transportation Systems held in Hawaii, USA, 4–7 November 2018.

**Abstract:** In this paper, we propose an advanced adaptive cruise control to evaluate the collision risk between adjacent vehicles and adjust the distance between them seeking to improve driving safety. As a solution for preventing crashes, an autopilot vehicle has been considered. In the near future, the technique to forecast dangerous situations and automatically adjust the speed to prevent a collision can be implemented to a real vehicle. We have attempted to realize the technique to predict the future positions of adjacent vehicles. Several previous studies have investigated similar approaches; however, these studies ignored the individual characteristics of drivers and changes in driving conditions, even though the prediction performance largely depends on these characteristics. The proposed method allows estimating the operation characteristics of each driver and applying the estimated results to obtain the trajectory prediction. Then, the collision risk is evaluated based on such prediction. A novel advanced adaptive cruise control, proposed in this paper, adjusts its speed and distance from adjacent vehicles accordingly to minimize the collision risk in advance. In evaluation using real traffic data, the proposed method detected lane changes with 99.2% and achieved trajectory prediction error of 0.065 m, on average. In addition, it was demonstrated that almost 35% of the collision risk can be decreased by applying the proposed method compared to that of human drivers.

**Keywords:** autonomous driving; adaptive cruise control; operation characteristic estimation; trajectory prediction

---

## 1. Introduction

According to the conducted survey, human errors have caused over 90% of car crashes [1]. To solve this problem, autonomous driving has been introduced as a solution that could substitute or help human drivers. However, the coexistence of human drivers and autonomous vehicles needs to be considered as a critical issue, as it is impossible to substitute human drivers at once completely. In the real-world, where people and automated machines coexist, understanding the operation characteristics of human drivers and predicting their future behavior are considerably important tasks to establish safe autonomous driving. Furthermore, at some point it will realize to forecast

dangerous situations and automatically generate maneuvers to prevent a collision. Currently, there are aggressive drivers conducting a risky lane change even when sufficient speed and distance are not guaranteed. It was reported that car crashes mainly occur because of lane changing [2]. In this case, if an autonomous vehicle suddenly decelerates to maintain the distance from a lane-changing vehicle, and it may lead to a crash with a following vehicle as shown in Figure 1. To prevent rear collisions caused by the cut-in situation, anticipation of future maneuvers of the surrounding vehicles positively necessary. Furthermore, automated control performed to avoid collisions based on the anticipation can significantly ensure driving safety.

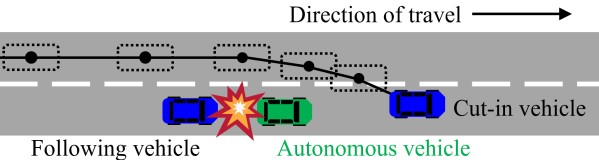

**Figure 1.** Rear collision caused by the interrupting vehicle: if the autonomous vehicle suddenly decelerates to avoid collisions with interrupting vehicles, it may lead to a crash with the following vehicle.

Two core techniques can be implemented to establish safe autonomous driving: First, trajectory prediction of surrounding vehicles is required. If future positions of adjacent vehicles can be predicted based on the registration of real-time movements, the autonomous vehicle will be able to generate safe paths to avoid possible collisions. Various methods have been proposed to predict the vehicle movements [3–7], and the models can be classified to macroscopic and microscopic models. Macroscopic model treats a numerous number of vehicles as flowing in a stream [6]. Although macroscopic model is effective to analyze the traffic flow, such as congestion or traffic volume, it is not appropriate for the collision avoidance system. On the other hand, microscopic model simulates the behavior of an individual vehicle, and it can be applied to predict future actions of adjacent vehicles [3–5,7]. However, they require the data on specific parameters to achieve the appropriate performance. In the aforementioned study, the values were statistically determined using the training data. However, the constant values cannot handle the individual difference as drivers generally have different operation characteristics. For instance, drivers demonstrate the different reaction time, which is one of the key operation characteristics depending on various factors such as age, driving experience, and gender [8]. Higgs et al. reported the limitation of the Wiedemann car-following model, which is one of the microscopic models, caused by the individual differences of drivers [7]. Previously suggested methods, including the ones suggested in our previous work, do not consider the individual difference in characteristics [9], leading to the loss of the trajectory prediction accuracy. In addition, the methods with the predetermined values proposed in the previous studies cannot handle the changes in driving conditions. If the traffic situation or environment is different from that implied within the training data, it also leads to deterioration of the overall performance. In [10], it is shown that the values of the parameters in the General Motors (GM) model vary over the training data.

To mitigate this problem, the real-time estimation of the operation characteristics is required as the second core technique. The proposed approach implies estimating the operation characteristics of a driver and applying the result to the trajectory prediction. You et al. proposed a method based on the extended Kalman filter to estimate the operation characteristics [11]. This method employs twelve parameters to model the individual characteristics of a driver. Filev et al. considered a driver behavior as the second order system [12]. This study estimated two parameters to represent the operation characteristics. Although the above two methods focused on estimation of the operation characteristics, they do not discuss the trajectory prediction. Note that Zhu et al. proposed a method to predict a trajectory using the deep learning framework [13]. Based on real experiments, it was proven that the method achieves higher prediction accuracy compared to the previously proposed ones. However, the data-driven approach has the limitation that the performance largely depends on

the chosen training dataset. If the real conditions differ significantly from that of the training dataset, the performance of the approach may deteriorate.

Considering the above limitations of previous studies, we propose a novel method to anticipate maneuvers of adjacent vehicles and to prevent a collision with them. Adaptive cruise control (ACC), which adjusts the speed and distance from the preceding vehicle, is already commercialized in recent years. Treiber et al. proposed the intelligent driver model (IDM) for the ACC system [14]. This method calculates the acceleration of the ACC vehicle based on the preferences of host driver, which are the desired speed, the desired time gap, and the desired deceleration with respect to the preceding vehicle. However, Kesting et al. reported that the IDM shows unrealistic behavior when the cut-in vehicle exists [15]. Davis proposed a method to consider not only the preceding vehicle but also the cut-in vehicle near an on-ramp [16]. However, this method assumes only mandatory lane changes performed near an on-ramp or off-ramp. Therefore, it is limited to handle discretionary lane changes on freeways. Milanés et al. proposed the ACC and cooperative ACC (CACC), which uses a vehicle-to-vehicle (V2V) communications system [17]. They achieved smooth and stable car-following behavior by applying the CACC. Lu et al. proposed the smart driver model (SDM) to improve the traffic flow stability [18]. By simulation, it was demonstrated that the SDM is able to stabilize the traffic flow under cut-in condition compared to previous models. However, the authors of [17] and [18] assume that all adjacent vehicles, including the host vehicle, implement the ACC and there is no manual driving vehicle. Therefore, the effectiveness of the two methods cannot be guaranteed under the condition where human drivers and autonomous vehicles coexist. In the simulation conducted by [18], the sudden deceleration was confirmed when a cut-in occurred. As human drivers cannot respond the sudden deceleration unlike the ACC vehicles, rear collision may occur. According to the previous study, ACC can occasionally lead to a rear collision, as it only focuses on the preceding vehicle [19]. By contrast, the method proposed in the present paper predicts the trajectories of the surrounding vehicles, including not only lane-changing vehicles, but also the following ones, behind the autonomous vehicle in question. In addition, the proposed method estimates the operation characteristics of other drivers and applies these estimates to the trajectory prediction. Then, the method assesses the collision risk according to the predicted trajectories and automatically controls its speed to maintain a safe distance from both the lane-changing and the following vehicles to minimize a risk indicator. Through the dynamic characteristic potential field method, which determines the distribution of potential fields depending on the relative distance and speed [20], the risk index is evaluated in this paper. This index has no restricting specific condition, as observed in the time-to-collision (TTC) when the collision risk is evaluated. Then, the proposed method finds an optimal position between the preceding and following vehicles to minimize the risk index.

The contribution of this paper is as follows. The proposed method performs the real-time estimation of operation characteristics of each driver while previous studies determine the constant values of model parameters from the training data. Our approach concludes the improvement of trajectory prediction accuracy and driving safety. A driver is aware of the distance and relative speed with respect to the preceding vehicle; consequently, the driver determines its acceleration as a response. The proposed method uses the GM model to analyze the behavior of the following driver, as it is widely employed among available car-following models [10]. The model has three parameters to represent the operation characteristics. Several studies reported the optimized values of the parameters using real traffic datasets [21–23]. However, the values vary across the papers, as all of the data was recorded at different locations. Therefore, the constant values of the parameters cannot be adjusted according to the changes in the driving conditions. To address this limitation, the proposed approach aims to estimate the real-time values of the parameters using the Levenberg–Marquardt algorithm [24,25]. This approach considers the changes in the driving conditions, unlike the previous methods based on the predetermined values. Moreover, the proposed method estimates the reaction time of the following driver in real time. The previous methods used the fixed reaction time of 1 s, although the performance of the trajectory prediction largely depends on the reaction time [13]. In this paper,

the three parameters of the GM model and the reaction time are defined as the operation characteristic variables. The proposed method performs the optimization of the four considered variables and applies the result to the trajectory prediction at each time step. Owing to this approach, the trajectory prediction can be robust with respect to the individual difference between the operation characteristics of drivers and the changes in driving conditions. Consequently, the anticipation of the future risk, with a high level of precision, and the collision avoidance can be realized. This is a state-of-the-art approach, and it is expected to be applied to an advanced safety system.

This paper is organized as follows. Section 2 describes the problem definition and a schematic of the proposed method. Section 3 explains the details of the proposed method. Section 4 presents the experiments and results. Section 5 describes the discussions including areas of future work. Finally, Section 6 describes the conclusions of this paper.

## 2. Overview

### 2.1. Problem Definition

This paper assumes the driving condition in which human drivers and autonomous vehicles coexist. Figure 2 represents the driving scene assumed in this paper. There are human drivers around the autonomous vehicle, which is equipped with embedded measurement devices, such as a GPS tracker and laser scanners used to acquire the data on movements of surrounding vehicles. The sensing range is assumed to be within 120 m. The relative speed and distance of adjacent vehicles can be acquired. The proposed method is implemented in the autonomous vehicle.

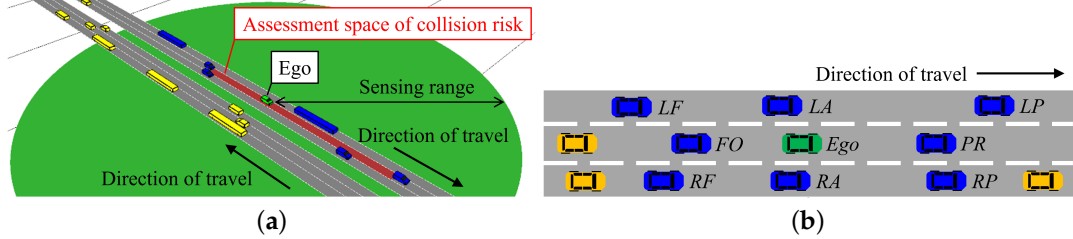

(**a**)  (**b**)

**Figure 2.** Problem definition: the green vehicle represents the autonomous vehicle, and the blue ones are the adjacent vehicles, which are the targets of the proposed method. The yellow vehicles are not considered in the proposed method. (**a**) The autonomous vehicle has measurement devices used to acquire the data on the distance and speed of the adjacent vehicles. (**b**) The autonomous vehicle is defined as ego, and the maximum number of considered adjacent vehicles is eight.

This paper assumes a situation when one of adjacent vehicles cuts in the front space of the autonomous vehicle, therefore reproducing one of the main factors of a crash. The future position of the cut-in vehicle for a time horizon of 2 s is predicted, and the collision risk is derived from the prediction result. The collision risks with respect to not only the cut-in vehicle, but also to the following vehicle, should be considered. If the autonomous vehicle immediately decelerates to maintain a distance from the cut-in vehicle, the rear collision can occur. Therefore, the risk assessment towards the two vehicles is strongly required.

In this paper, the autonomous vehicle is defined as ego, and eight adjacent vehicles are defined, as shown in Figure 2b. The ego vehicle is indicated using green color, and the eight adjacent vehicles are depicted using blue color. The yellow vehicles represent non-target vehicles out of the coverage of the proposed method. The maximum number of considered adjacent vehicles is eight. In the figure, LF represents the following vehicle on the left lane, LA is the alongside vehicle on the left lane, and LP denotes the preceding vehicle on the left lane. FO is the following vehicle on the same lane of the ego vehicle; and PR represents the preceding vehicle of the ego vehicle. In the same way, RF represents the following vehicle on the right lane; RA is the alongside vehicle on the right lane, and RP denotes the

preceding vehicle on the right lane. The ego vehicle is always monitoring the adjacent vehicles while estimating their driving intentions and predicting their trajectories.

### 2.2. Overview of the Proposed Method

An advanced ACC (AACC) system is proposed based on the operation characteristic estimation and trajectory prediction to solve the limitations described in Section 1. Figure 3 shows the schematic of the proposed method. The AACC consists of two parts: a predictor and a planner. The predictor is generated according to the number of adjacent vehicles. If there are $N$ adjacent vehicles around the ego vehicle, $N$ units are generated, and then each unit predicts the trajectory of one adjacent vehicle. The inputs of the predictor are the data on the position and speed of the ego and the adjacent vehicles. These information can be acquired using GPS and a controller area network (CAN) bus. Laser scanners are installed to measure the position and speed of the adjacent vehicles. The predictor consists of three subparts: driving intention estimation, operation characteristic estimation, and trajectory prediction. The outputs of the predictor are the trajectories of $N$ adjacent vehicles.

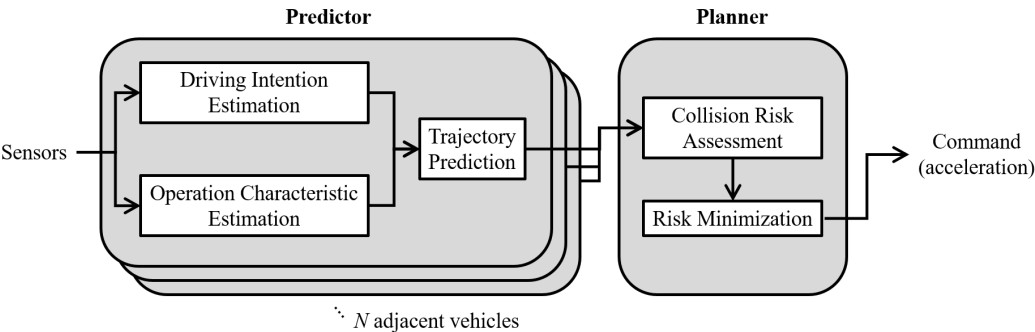

**Figure 3.** Schematic of the proposed method: the green vehicle represents the ego vehicle, and the blue ones are the adjacent vehicles, which are the targets of operation characteristic estimation and trajectory prediction. The ego vehicle is equipped with measurement devices used to acquire the data on the distance and speed of the adjacent vehicles. The proposed method is implemented within the ego vehicle.

First, this paper defines four driving intentions, "keeping", "changing", "arrival", and "adjustment". The proposed method handle the intention estimation as a multiclass problem. Each intention indicates a class, and it is estimated by the support vector machine (SVM). The lateral movement of the adjacent vehicles is used as a feature. Details of this method are provided in Section 3.1. The output of this part is the driving intention of each driver.

Second, the operation characteristics of the adjacent drivers are estimated at each time step. Using the measured information, four operation characteristic variables are estimated: three parameters of the GM model and the reaction time. The proposed method applies the Levenberg–Marquardt algorithm to determine the real-time values of the considered variables. The details of this part are explained in Section 3.2.

Third, the method applies the estimated driving intention and operation characteristic variables to perform the trajectory prediction. In general, different schemes are conducted according to different intentions. When drivers intend to keep the current lane such as keeping and adjustment, they aim to maintain the safe distance from the preceding vehicle of the same lane. However, when drivers intend to change a lane such as changing and arrival, they must consider the vehicles in the adjacent lane. The proposed method is based on this assumption. Two prediction methods are applied to each direction. For the longitudinal direction, the GM model is used to calculate acceleration based on the estimated values of operation characteristic variables. For the lateral direction, the proposed method uses the sinusoidal model based on the estimated intention. Section 3.3 describes details of the trajectory prediction method.

Based on the outputs of the predictor, the planner calculates the acceleration of the ego vehicle to maintain the safe distance from adjacent vehicles. The planner consists of two subparts: collision risk assessment and risk minimization. The dynamic potential field method is applied to evaluate the collision risk [20]. As the risk index, the repulsive potential energy generated from the adjacent vehicle is defined. The large amount of the energy is produced when two vehicles are near each other and move rapidly. In contrast, the small repulsive potential energy is produced when the ego vehicle maintains a sufficient distance from the adjacent vehicle and adjusts its speed appropriately. The value of repulsive energy reflects the collision risk. Section 3.4 explains details of this approach.

Finally, the proposed method finds an optimal position to minimize the risk index. In this phase, the collision risks with respect to both the cut-in vehicle and following vehicle of the ego vehicle are considered. Moreover, inconsistent or excessive acceleration (deceleration) need to be strictly controlled, as such actions can cause the rear collision. The output of this part is the control value for the ego vehicle. Section 3.5 presents details of this part.

## 3. Proposed Method

### 3.1. Driving Intention Estimation

It can be assumed that all drivers simply have two intentions: lane-keeping and lane-changing. When a driver is satisfied with a current driving condition, he/she may keep a current lane and only focus on maintaining a safe distance from the preceding and following vehicles. However, if the driver is in a state of dissatisfaction with the current driving condition, he or she may try to change a lane. For the trajectory prediction, this paper defines the driving intentions more specifically as shown in Figure 4.

When a driver has an intention of keeping, the driver control the speed for maintaining a safe distance from the preceding vehicles. By contrast, changing expresses an intention to start changing a lane until crossing the lane marking. The intention of arrival represents a step until the vehicle reaches the center of the target lane, after it is over the lane marking. When a driver has an intention of adjustment, drivers start to adjust the speed with respect to vehicles on the target lane.

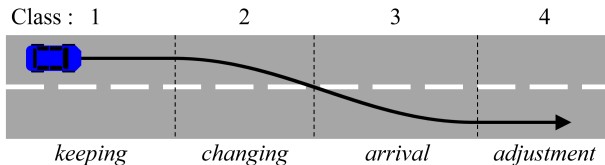

**Figure 4.** Definition of driving intentions: driving intentions may correspond to four classes: keeping, changing, arrival, and adjustment.

The proposed method defines each intention as a class and treats the estimation of driving intentions as a multiclass problem using the SVM. Two features representing the lateral movement of the target vehicle are extracted: the distance from the centerline and the lateral speed. In addition, one feature is extracted to describe the driving condition around the target vehicle. Details on how to extract the features are described in our previous work [26]. The feature vector at time $t$ can be represented as follows.

$$\mathbf{x}_t^{(k)} = \left[ \mathbf{d}_t^{(k)} \ \dot{\mathbf{d}}_t^{(k)} \ \mathbf{p}_t^{(k)} \right]^{\mathrm{T}}, \tag{1}$$

$$\mathbf{d}_t^{(k)} = \left[ d_{t-(W-1)}^{(k)} \ \cdots \ d_{t-1}^{(k)} \ d_t^{(k)} \right], \tag{2}$$

$$\dot{\mathbf{d}}_t^{(k)} = \left[ \dot{d}_{t-(W-1)}^{(k)} \ \cdots \ \dot{d}_{t-1}^{(k)} \ \dot{d}_t^{(k)} \right], \tag{3}$$

$$\mathbf{p}_t^{(k)} = \left[ p_{t-(W-1)}^{(k)} \quad \cdots \quad p_{t-1}^{(k)} \quad p_t^{(k)} \right], \tag{4}$$

where $k$ is the index that denotes the lane marking. Both left and right lane-changing can be adapted. Depending on the lane-changing side, all features are extracted by the specified side. Moreover, a moving window is set, as the lane-changing is a continuous process. $W$ denotes the size of moving window in Equations (2)–(4), and it is a parameter to determine the sequence memory for the continuous process. For instance, the distance from the centerline, $\mathbf{d}_t^{(k)}$, is a sequence that consists of the $W$ data until the time $t$.

The SVM kernel converts features from a low-dimensional space into a high-dimensional space to handle the complexity of driving intentions, which may lie in a high-dimensional feature space. The proposed method uses the radial basis function known to provide the best performance. A one-versus-one strategy is implemented in the proposed method for the multiclass classification. Finally, the intention at the current time step of each adjacent driver is the output in this part.

*3.2. Operation Characteristic Estimation*

The proposed method considers a driver's behavior as the stimulus–response system; consequently, the acceleration or deceleration is derived from the distance and relative speed with respect to the preceding vehicle of that vehicle. Among the various methods to model the lane-keeping movement, the proposed method employs the GM model, as numerous studies have been conducted to calibrate the GM model as representing human-like car-following behavior. Thanks to previous works, the GM model is considered as one of the best models, and the effectiveness was demonstrated by field experiments [3,27]. At any time step, $t$, let the longitudinal position of the corresponding vehicle be represented by $x_i^t$. Here, $i$ is an index of adjacent vehicles, and $i+1$ represents the preceding vehicle of the vehicle $i$.

$$\hat{a}_i^t = \left[ \frac{\alpha_{l,m}(v_i^t)^m}{(x_{i+1}^{t-\Delta T} - x_i^{t-\Delta T})^l} \right] (v_{i+1}^{t-\Delta T} - v_i^{t-\Delta T}), \tag{5}$$

where $x_{i+1}^t$ indicates the position of the vehicle $i+1$, and $v_{i+1}^t$ denotes its speed at time step $t$. Similarly, $x_i^t$ represents the position of the vehicle $i$ in the operation characteristic estimation, and $v_i^t$ corresponds to its speed at time step $t$. $\alpha$, $l$, and $m$ are the model parameters to determine the operation characteristics, and $\Delta T$ is the reaction time. These four variables are considered as the operation characteristic variables in this paper.

The proposed method performs the optimization of the three model parameters using the Levenberg–Marquardt algorithm [24,25]. Although there are many iterative optimization algorithms, such as the gradient descent or the Newton method, the Levenberg–Marquardt algorithm is generally used to solve nonlinear problems. Employing this algorithm, the proposed method estimates the optimized values at the current time based on information on the acceleration at the previous step. To perform optimization of the three model parameters, the reaction time is used at the previous step. If optimization fails, or the derived acceleration value is too large or small, the value of the previous step is used as a fallback value.

After optimization of the three model parameters ($\alpha$, $l$, and $m$), the estimation of the reaction time is performed. According to the previous study, the reaction time is distributed in the range between 0.92 and 1.94 s [8]. Including the room for distribution, the proposed method identifies the optimal value of the reaction time in the range from 0.5 to 2.5 s with increments of 0.1 s. The value can be derived as follows,

$$\Delta T = \arg\min_{\Delta T} |a_i^{t-1} - \hat{a}_i^{t-1}(\Delta T)|, \tag{6}$$

where $a_i^{t-1}$ represents the ground truth of acceleration and $\hat{a}_i^{t-1}$ denotes the derived value obtained using the proposed method.

Estimation of the operation characteristic variables is performed following the above process, and the optimal values are derived at each time step. However, the operation characteristics may not drastically change in a short time period. Therefore, the proposed method defines a sliding window of a constant size. Consequently, the values within the window are modified according to a moving average.

### 3.3. Trajectory Prediction

The future maneuver of adjacent vehicle is predicted according to the estimated driving intention of the vehicle. When keeping is estimated as the current intention, the lateral position is determined at the center of the current lane. On the other hand, the lateral position is set to the center of the adjacent lane when adjustment is estimated. If the estimated intention is changing or arrival, the lateral position of the target vehicle is predicted through the sinusoidal model [28]. Although numbers of lane change models have been proposed, the common models are isokinetic migration model, arc model, trapezoidal acceleration model, and sinusoidal model [29]. The isokinetic migration model is simple and easy to calculate; however, the generated maneuver would be unrealistic. The arc and trapezoidal acceleration models have the poor flexibility since they require to determine planning parameters. On the other hand, the sinusoidal model determines the lateral acceleration according to the duration of lane-changing, and the duration can be calculated without any particular parameters. The proposed method derives the duration using the lateral speed at the moment, when the lane-changing is detected, and the lane width. This model generates a trajectory, such as a sine curve, as shown in Figure 5. The acceleration in the lateral direction can be derived as follows,

$$a_{i,lat}^{t_c} = \frac{2\pi H}{t_{lat}^2} \sin \frac{2\pi}{t_{lat}} t_c, \tag{7}$$

where $a_{i,lat}^t$ indicates the lateral acceleration of the vehicle $i$, $t_c$ is the time from the beginning of lane departure, $H$ is the lane width, and $t_{lat}$ is the lane-changing duration. $t_{lat}$ is calculated using the lateral speed at the moment which the intention is estimated as changing. Therefore, the calculation of lateral acceleration does not require any particular parameters.

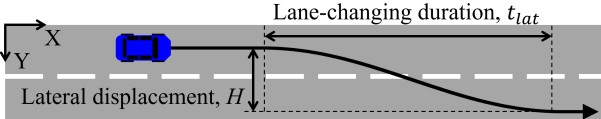

**Figure 5.** Sinusoidal model for lane-changing trajectory: the proposed method uses the sinusoidal model to predict the lateral movement, when the target vehicle performs a lane change. In the figure, $H$ is the final lateral displacement and $t_{lat}$ is the lane-changing duration.

The longitudinal position of adjacent vehicle is predicted based on the GM model, and the acceleration value using the estimated values of operation characteristic variables is calculated. Then, the position and speed of the vehicle $i$ are updated as follows.

$$\hat{v}_{i,lon}^{t+1} = v_{i,lon}^t + \hat{a}_{i,lon}^t \Delta t, \tag{8}$$

$$\hat{x}_{i,lon}^{t+1} = x_{i,lon}^t + \hat{v}_{i,lon}^t \Delta t, \tag{9}$$

where $\hat{x}_{i,lon}^t$ denotes the longitudinal acceleration derived by Equation (5). Based on the estimated driving intention, the proposed method identifies the preceding vehicle for the target vehicle to follow. When the estimated intention is keeping, the driver follows the preceding vehicle of the current lane. Then, if the driver has the intention of changing, arrival, or adjustment, he/she may aim to follow the preceding vehicle of the adjacent lane. The proposed approach reflects such tendency of drivers,

and the preceding vehicle is selected as shown in Figure 6. It is assumed that the driver keeps the current acceleration until the trajectory prediction is over when there is no preceding vehicle.

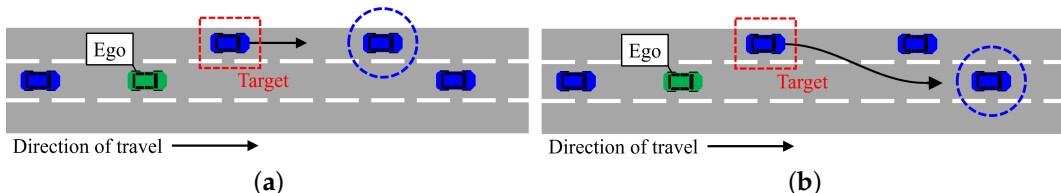

(**a**)　　　　　　　　　　　　　　　　　　　　　　(**b**)

**Figure 6.** Vehicle selection for longitudinal prediction: when the trajectory prediction is performed for the target vehicle surrounded by a red rectangle, the preceding vehicle to follow is chosen. (**a**) If the target driver has the intention of keeping, the driver aims to follow the preceding vehicle of the current lane surrounded by a blue circle. (**b**) By contrast, if the driver has the intention of changing, arrival, or adjustment, the preceding vehicle of the adjacent lane is selected.

The trajectory for the time horizon of two seconds is predicted since the reaction times of drivers is commonly reported within two seconds [8]. Consequently, the sequence of future positions of the adjacent vehicles for two seconds in advance is generated.

### 3.4. Collision Risk Assessment

The collision risk is assessed based on the predicted future positions. As described in Section 2.2, the dynamic potential energy is defined as the risk index. The repulsive potential energy from the vehicle $i$ is derived by

$$f(\Delta V_i, \theta_i) = \frac{1}{2\pi I_0(\eta(\Delta V_i))} \exp\left(\eta(\Delta V_i)\cos\theta_i\right), \tag{10}$$

$$h(G_i) = \frac{1}{2\pi\sigma_i} \exp\left(-\frac{G_i^2}{2\sigma^2}\right), \tag{11}$$

$$R_i = f(\Delta V_i, \theta_i)h(G_i), \tag{12}$$

where

$$\theta_i = \begin{cases} \pi & (i = PR) \\ 0 & (i = FO) \end{cases}. \tag{13}$$

Here, $\Delta V_i$ represents the relative speed between the ego and vehicle $i$, and $G_i$ is the distance from the vehicle $i$. The von Mises distribution is applied in Equation (10), and $I_0(\eta)$ represents a modified Bessel function of order 0. If the parameter $\eta$ is zero, the uniform filed is generated. When the parameter $\eta$ is not zero, the drifted distribution toward the angle $\theta_i$ is generated. The value of $\eta$ determines the drifted amount of distribution, and it is derived from the relative speed $\Delta V_i$.

Figure 7 shows the distribution of the generated potential field according to the relative speed between the ego vehicle and the vehicle $i$. The colors of the potential field indicate the energy level. The red color depicts the high energy level, and the blue represents the low level. When the ego and the vehicle $i$ show the same speed, the uniform distribution is generated as shown in Figure 7a. If the vehicle $i$ moves slower than the ego vehicle, the distribution with bias toward the ego vehicle is generated as shown in Figure 7b. Consequently, the large potential energy implies the ego vehicle, and it represents the high risk of colliding with the vehicle ahead. On the other hand, if the vehicle $i$ is faster than the ego vehicle, the potential field is drifted to the forward direction as shown in Figure 7c. Although the ego vehicle drives close to the vehicle $i$, the collision risk is assessed as low because of the relative speed. In this case, the small amount of the repulsive potential energy indicates the low collision risk. Equation (11) defines the repulsive potential energy generated from the vehicle $i$, and the value is inversely proportional to the distance between the two vehicles. This equation represents that

the potential energy is lower when the ego vehicle is farther away. In contrast, if the two vehicles are close, the large amount of energy affects to the ego vehicle.

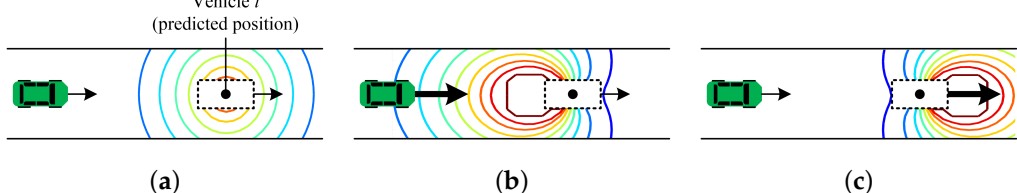

**Figure 7.** The distribution of potential field: (**a**) The ego and the vehicle *i* drive with the same speed, (**b**) the ego is faster than the vehicle *i*, and (**c**) the ego is slower than the vehicle *i*. As shown, the potential field is distributed depending on the relative speed between the two vehicles.

The collision risks towards both the preceding and following vehicles are assessed. When the cut-in occurs in the front space of the ego vehicle, the collision risk with respect to the cut-in vehicle is assessed instead of considering the preceding vehicle of the current lane. Figure 8 represents the way how the assessment space of the collision risk is determined by the existence of cut-in vehicles. Finally, the collision risk at position $(x, y)$ is calculated by

$$R(x, y) = \sum_{i=PR,FO} R_i. \tag{14}$$

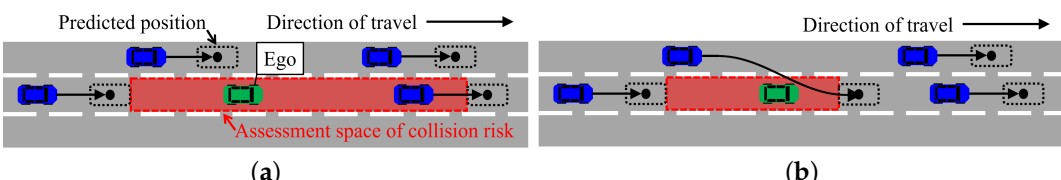

**Figure 8.** Definition of the assessment space: (**a**) When there is no lane-changing vehicles around the ego, the assessment space is defined between the preceding and following vehicles in the same lane. (**b**) If the adjacent vehicle is predicted to change a lane to the front space of the ego vehicle, the assessment space is determined between the predicted position of the lane-changing vehicle and that of the following vehicle.

### 3.5. Risk Minimization

When there is no cut-in vehicle, the ego vehicle considers the collision risks to both the preceding and following vehicles. On the other hand, if the lane-changing is detected, the ego vehicle references the cut-in vehicle based on the predicted lane-changing maneuver. The proposed method finds the optimal position between the two vehicles to minimize the collision risk as shown in Figure 9. To avoid a collision with the cut-in vehicle, if the ego vehicle conducts inconsistent or excessive acceleration (deceleration), it can conversely cause a crash, as it is unexpected for the following driver. Thus, the acceleration (deceleration) is limited within $\pm 0.5$ m/s². The optimal position to minimize the risk index is derived by

$$(x^*, y^*) = \arg\min_{x,y} R(x, y), \tag{15}$$

where

$$x_f < x < x_p. \tag{16}$$

In Equation (16), $x_f$ indicates the position of the following vehicle and $x_p$ represents that of the preceding vehicle. Finally, the control value is derived, as the ego vehicle arrives at the position $(x^*, y^*)$ two seconds in advance.

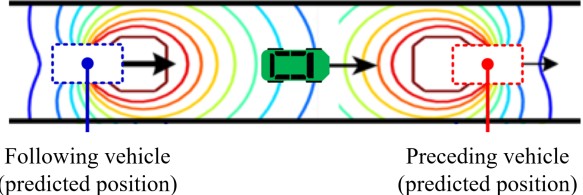

Following vehicle
(predicted position)

Preceding vehicle
(predicted position)

**Figure 9.** Risk minimization: if there is no cut-in vehicle, the advanced adaptive cruise control (AACC) assesses the collision risks with respect to the preceding and following vehicles. Otherwise, the AACC assesses the collision risk generated from the cut-in vehicle based on the predicted lane-changing maneuver when the cut-in occurs.

## 4. Evaluation

### 4.1. Dataset

For evaluation of the proposed method, real traffic data were used for the analysis [30]. The traffic flow on a highway in Germany was recorded by a drone. The data were gathered at six locations, and the time series data for the sets of 110 and 500 vehicles were included. The position, speed, acceleration, size, and other parameters of each vehicle were described. A 4K camera was implemented within the drone, and the measurement accuracy was approximately 10 cm. The measurement rate was 25 Hz. The highway at location 1 has two lanes per direction, and the other locations have three lanes per direction. To validate the robustness of the proposed method with respect to driving conditions, the performance was evaluated using the data from all locations. For the evaluation, 5917 lane-keeping vehicles and 1010 lane-changing vehicles were considered.

### 4.2. Performance of the Driving Intention Estimation

The estimation accuracy of driving intention was measured. The cases were counted as a failure when the proposed system anticipated the lane-keeping, when in fact the vehicle conducted a lane-changing. On the other hand, the cases were counted as a false alarm when the proposed method judged that the adjacent vehicle would perform a lane change, when in fact the lane-changing did not occur. Although it is equally important to decrease the number of false alarms, the failure is the most dangerous case. Consequently, a recall with 100% accuracy must be achieved for the safety system. The evaluation was conducted using the $F_1$ score, which is derived by

$$F_1 = 2 \times \frac{\text{precision} \times \text{recall}}{\text{precision} + \text{recall}}. \tag{17}$$

The precision represents the false-alarm rate, and the recall evaluates the failure rate. Through evaluation using the entire testing data, the $F_1$ score of 99.2% was achieved with the proposed method. False alarms occurred in 95 cases among 5917 lane-keeping events. Otherwise, no failures occurred among 1010 lane-changing events. It was confirmed that the proposed method satisfies the requirement without failures and achieves the great accuracy with the $F_1$ score of 99.2%.

Figure 10 illustrates an example of the driving intention estimation. In the figure, $\tau_j$ indicates the time of detecting the lane-changing of adjacent vehicle, and $\tau_c$ represents the time of crossing the lane marking by the vehicle. It is confirmed that the lane-changing was successfully detected in advance before crossing the lane marking.

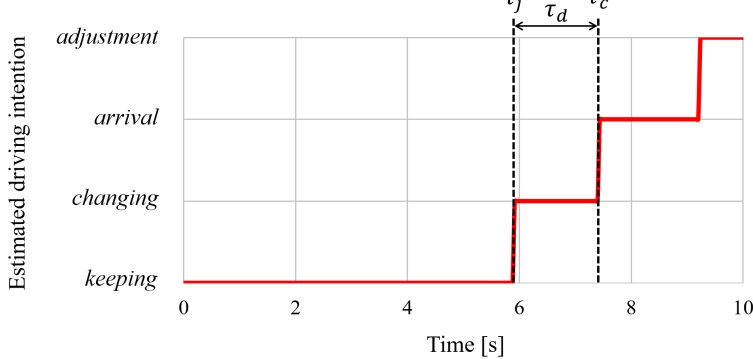

**Figure 10.** Example of the driving intention estimation: In the figure, $\tau_j$ indicates the time of detecting the lane-changing of adjacent vehicle, and $\tau_c$ represents the time of crossing the lane marking by the vehicle. It is confirmed that the lane-changing was successfully detected in advance before crossing the lane marking.

The system should detect a lane change as soon as possible, at the moment the lane changing maneuver starts. The detection speed $\tau_d$ can be defined as follows,

$$\tau_d = \tau_c - \tau_j. \tag{18}$$

A large value of $\tau_d$ implies that the ego vehicle acquires a sufficient time to react, improving driving safety. If the detection is delayed, the ego vehicle cannot be ready to react to the interruption of adjacent vehicles. Consequently, the accuracy and the detection speed of the driving intention estimation are critical factors determining the performance of the proposed system. However, there is a trade-off between accuracy and detection speed [31]. To overcome the limitation, our research group has tried to make a breakthrough, and one of considered approaches is described in our previous work [32]. Details on this point are explained in Section 5.

*4.3. Results of the Operation Characteristic Estimation*

Figure 11 shows the result for a driver of the adjacent lane from the ego vehicle. The X-axis depicts the time, and the Y-axis represents the estimated values of operation characteristic variables. The blue line indicates $\alpha$, the green one is $l$, and the black line shows $m$. These parameters do not have units. It is confirmed that stable values were estimated for all parameters. In addition, the red line indicates the reaction time (in seconds). In the figure, although the reaction time is slightly unstable, the value is distributed around 1.5 s. This value is in the range between 0.92 and 1.94 s, as reported in the previous study [8]. As mentioned in Section 3.2, the operation characteristics may not drastically change in a short time period including the reaction time. Therefore, the moving average window was set with the constant size of 1 s. However, the value was manually determined, and the large value may be more appropriately from this result. The specified parameter setting needs to be investigated as a part of the future work.

The estimated value at each time is applied to the trajectory prediction; then, the acceleration towards adjacent vehicles can be calculated based on the real-time estimation. As there is no ground truth for the operation characteristic variables, it is impossible to evaluate the performance of the proposed method in this case. The effectiveness of the real-time estimation is discussed in Section 4.4.

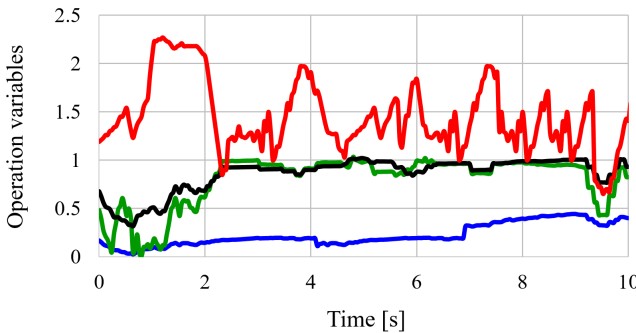

**Figure 11.** Examples of the operation characteristic estimation using the proposed method: the blue line indicates *α*, the green one is *l*, and the black line shows *m*. These parameters do not have units. In addition, the red line indicates the reaction time (in seconds). It is confirmed that the reaction time is distributed around 1.5 s.

### 4.4. Performance of Trajectory Prediction

The error between the ground truth and the predicted position of the adjacent vehicles was considered as the evaluation criterion of trajectory prediction. Considering that the proposed method predicts the future positions until 2 s with increments of 0.04 s, the root-mean-squared error (RSME) of all predicted positions was considered as the criterion. Let *n* be the index of the predicted position, and then the RMSE can be calculated as follows,

$$\text{RMSE} = \sqrt{\frac{1}{N}\Sigma_{n=1}^{N}[(x_n - \hat{x}_n)^2 + (y_n - \hat{y}_n)^2]}, \tag{19}$$

where $x_n$ and $y_n$ denote the position at the incremental step *n*, which is used as the ground truth. $\hat{x}_n$ and $\hat{y}_n$ represent the predicted position obtained using the proposed method. However, the error of lateral positions was excluded from the evaluation scope. *N* is the number of predicted positions. The future positions are predicted until two seconds with increments of 0.04 s; therefore, *N* is equal to 50.

Figure 12 shows an example of the predicted future positions of the adjacent vehicles obtained using the proposed method. The green vehicle indicates the ego vehicle, whereas the blue ones represent the adjacent vehicles for which the ego vehicle is monitoring intentions and behavior. The blue rectangles show the predicted lane-keeping trajectories, and the red ones represent the lane-changing trajectory. The figure shows the future positions at five prediction terms: 0.4 s, 0.8 s, 1.2 s, 1.6 s, and 2 s. From this figure, it can be seen that the cut-in vehicle is detected, and the trajectory of that vehicle is successfully predicted. Consequently, it is possible to ensure sufficient response time for the ego vehicle based on the predicted future positions of the adjacent vehicles.

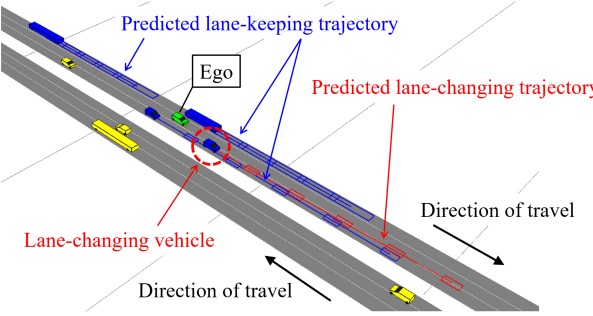

**Figure 12.** Example of trajectory prediction: the blue rectangles show the predicted lane-keeping trajectories, and the red ones represent the lane-changing trajectory. It is confirmed that the cut-in vehicle is detected, and the trajectory of that vehicle is successfully predicted.

To evaluate the effectiveness of the real-time estimation of operation characteristics, the prediction accuracy was compared to those of the previous methods that employ constant values of the operation characteristic variables, as shown in Table 1. As the previous methods cannot predict the lane-changing trajectory, only lane-keeping events were used for the performance comparison. It is clearly evident that the proposed method with the adjustment of operation characteristics significantly improves the prediction accuracy. In Table 1, the minimum error for each location is shown in bold. Compared to the results obtained using the previous methods, the proposed method considerably reduced the errors by almost the half of the number of errors reported for the previous method in [21]. The average error of the proposed method was 0.065 m, whereas that of [21] was 0.139 m.

**Table 1.** Comparison of the performance of the proposed method with those of the previous methods.

| Location ID | Heyes [21] | Ozaki [22] | Aron [23] | Proposed |
|:---:|:---:|:---:|:---:|:---:|
| 1 | 0.159 m | 0.215 m | 0.211 m | **0.065 m** |
| 2 | 0.130 m | 0.178 m | 0.181 m | **0.062 m** |
| 3 | 0.093 m | 0.150 m | 0.133 m | **0.055 m** |
| 4 | 0.115 m | 0.165 m | 0.148 m | **0.062 m** |
| 5 | 0.180 m | 0.216 m | 0.218 m | **0.079 m** |
| 6 | 0.155 m | 0.202 m | 0.195 m | **0.070 m** |
| Average | 0.139 m | 0.189 m | 0.183 m | **0.065 m** |
| Standard deviation | 0.032 m | 0.028 m | 0.034 m | **0.008 m** |

Considering the results of the three previous methods, it can be seen that the performance is largely affected by the values of the parameters. Among the previous methods, the values of the operation characteristic variables in [21] showed the best accuracy. However, it was confirmed that the average errors varied depending on the location. Therefore, it is concluded that the constant values cannot be adjusted according to the change in the driving conditions, which may lead to a deterioration of the performance. By contrast, the proposed method showed the robust performance in various driving conditions owing to the real time estimation of the parameter values. It was confirmed that the change in the location did not significantly impact the performance of the proposed method. The standard deviation of the proposed method was 0.008 m, whereas that of the method proposed by [21] was 0.032 m. Furthermore, the proposed method achieved the best accuracy at all locations compared to the previous methods. It means that the real-time estimation of operation characteristics can account for the individual difference in driver behavior and the change in driving conditions appropriately. Based on this comparison, it was demonstrated that the real-time optimization of the operation characteristic variables is significantly effective in improving the robustness of the trajectory prediction.

*4.5. Performance of Collision Risk Minimization*

Table 2 represents the average of the collision risk for the entire dataset. It can be confirmed from the table that the proposed system is significantly effective to decrease the collision risk compared to human drivers. Based on the predicted trajectory, the ego vehicle appropriately adjusted the distance and speed with respect to the adjacent vehicles. As a result, it considerably improved the driving safety with respect to that of human drivers. Figure 13 illustrates the speed of the ego vehicle under a cut-in situation. A human driver rapidly reduced the speed after the cut-in vehicle crossed the lane marking. Fortunately, a crash did not occur in this case; however, the unexpected deceleration was confirmed at ~7 s. Meanwhile, it was shown that the proposed system maintained the speed without the rapid deceleration since the lane-changing was anticipated in advance. It means that the rear collision can be prevented compared to human drivers. In addition, the two state-of-the-art ACC strategies were

implemented to compare the performance with the proposed method. The ACC system in [17] derives the acceleration of the ego vehicle by

$$a_{ACC} = k_1(x_p - x_e - t_{hw}v_e) + k_2(v_p - v_e), \tag{20}$$

where $x_p$ and $x_e$ represent the current position of the preceding vehicle and that of the ego vehicle, respectively; $v_p$ and $v_e$ represent the current speed of the preceding vehicle and that of the ego vehicle, respectively; $t_{hw}$ is the current time-gap between the two vehicles; $K_1$ and $k_2$ are the gains. Based on the setting in [17], the values of the two gains were determined as follows: $k_1 = 0.23$ and $k_2 = 0.07$. The SDM in [18] calculates the acceleration of the ego vehicle by

$$a_{SDM} = a_{max}\left[1 - \left(\frac{v_e}{v_0}\right)^\delta\right] - \frac{a_{max}\left[1 - \left(\frac{v_e}{v_0}\right)^\delta\right] + \frac{v_e^2 - v_p^2}{2(x_p - x_e)}}{\exp\left(\frac{x_p - x_e}{s_0 + v_e T} - 1\right)}, \tag{21}$$

where $a_{max}$ represents the maximum acceleration, $v_0$ is the maximum speed, $\delta$ is the acceleration exponent, and $s_0$ is the standstill distance between stopped vehicles; $T$ represents the desired time gap. The values of parameters were determined as follows, $a_{max} = 1.4$, $v_0 = 30$, $\delta = 4$, $s_0 = 1.5$, and $T = 1.6$, according to [18]. The AACC achieved the best performance to decrease the collision risk compared to the two previous methods. As the AACC considers not only the preceding vehicle, but also the following vehicle, the collision risk with respect to the following vehicle can be improved. Based on these results, it is demonstrated that the AACC allows improving the driving safety and outperforms both human drivers and the previous ACC systems.

**Table 2.** Comparison of collision risk.

|  | Human drivers | ACC [17] | SDM [18] | AACC |
| --- | --- | --- | --- | --- |
| Collision risk [J] | 1.21 | 0.89 | 0.90 | 0.79 |

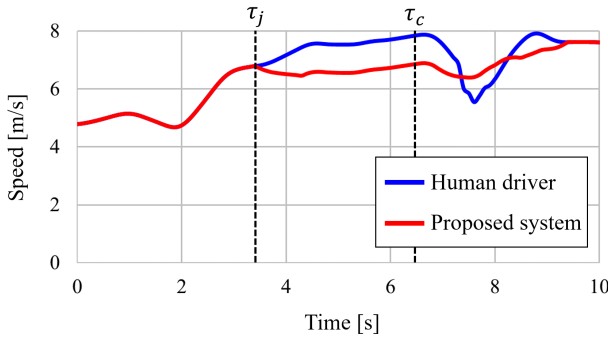

**Figure 13.** Comparison of the speed profile: the blue line represent the speed profile of a human driver, while the red line is that of the proposed system. Compared to the rapid deceleration performed by the human driver after the cut-in vehicle crossed the line marking, the proposed system maintained the speed without the rapid deceleration since the lane-changing was anticipated in advance.

## 5. Discussion

Based on the results discussed in Section 4.5, it was concluded that the proposed method was able to improve the driving safety compared to human drivers. Generally, drivers pay more attention to keep a safe distance and speed with respect to the preceding vehicle, rather than the following one. Therefore, this tendency may lead to a collision with the following vehicle, if the driver rapidly decreases the speed. By contrast, the proposed approach enables maintaining the safety with respect to both the preceding and following vehicles. Furthermore, the proposed method allows preventing the future collision risk owing to the trajectory prediction of adjacent vehicles around the ego vehicle.

The proposed approach maximizes the strength of automatic control to thoroughly monitor the surrounding conditions; consequently, it serves to guarantee the safety of driving.

As mentioned in Section 4.2, the performance of the proposed method in terms of the driving safety largely depends on the speed of the lane-change detection. If it is possible to anticipate a lane change much faster, sufficient response time to allow the ego vehicle adjusting the speed without rapid acceleration or deceleration is ensured. In our previous work [32], the aggressiveness of the driving mode was estimated in terms of lane-changing, and the estimation result was applied to the detection of lane changes. Consequently, our research group achieved the improvement of detection speed without the loss of detection accuracy. In this paper, despite the fact that the approach to estimate the aggressiveness of lane-changing is not discussed, it is assumed that there is the relationship between the aggressiveness of lane-changing and the operation characteristics in car-following. If that relationship is thoroughly analyzed, it contributes to drastically improving the performance of the proposed system. Furthermore, the machine learning technique was applied to estimate the aggressiveness of driving mode in the previous method. However, the proposed machine learning technique has the limitation with respect to changes in the driving conditions. A new approach to consider the aggressiveness of drivers and handle the change of driving conditions is required, and it is one of planned topics for our future works.

The real-time estimation of operation characteristics of each driver is considered as the main factor to improve the performance of trajectory prediction. The effectiveness of the proposed approach was confirmed based on the results of the comparison presented in Section 4.4. However, the unstable estimation was conducted in some cases. Based on the assumption that the operation characteristics do not drastically change in a short time period, the proposed method defines the moving window of the constant size for the operation characteristic estimation and obtains a moving average within the window. However, the unstable estimation results indicate that the moving window cannot cope with some cases, as the size of moving window was manually tuned. This point is also considered to be investigated in the future research work.

We have planned to implement the proposed method on a real vehicle and have installed the sensor system for measurement. The vehicle consists of a position sensor and six laser scanners [26]. The RT3003 is used as the position sensor and has update rate of 100 Hz. The laser scanner is an ibeo LUX that has an update rate of 32 Hz. The proposed method should satisfy the computation limit derived from the sensor system. Considering the capability of the equipment used, the computation of the proposed method should be finished within 30 ms. The whole calculation time of the proposed method was tested, and the specification of the testing machine is described as follows; Intel Core i7-8700, 3.20 GHz CPU. The maximum calculation time of the proposed method was 11 ms using the entire testing dataset. This means that the proposed method is able to satisfy the system requirement.

The proposed method has two limitations. The first limitation is caused by the assumption that the operation characteristics may not drastically change in a short time period. If the operation characteristics are drastically changed, the precision of trajectory prediction may degrade. The second limitation is that the proposed method cannot handle abnormal behaviors of adjacent vehicles. The proposed method assumes that a driver is in a normal state. If the drivers is in abnormal states, such as sleepiness or inattention, it may lead to deterioration of the overall performance. Therefore, the proposed system is for SAE level 3 automation. Although the driving is fully automated under normal conditions, the driver should be ready to substitute the operation. For level 4 or 5 automation, the development to detect abnormal drivers and anticipate their behaviors is required, and that point is our future work.

## 6. Conclusions

This paper proposed a novel advanced adaptive cruise control to improve driving safety through the anticipation of future maneuver of adjacent vehicles and collision risks with them. In particular, we focused on the cut-in situation, in which the surrounding vehicle intrudes into the front space of the

ego vehicle. The proposed system predicted maneuvers of adjacent vehicles based on the estimation of their intentions. For better prediction, the real-time estimation of operation characteristics of each driver was performed, and it allowed achieving the drastic improvement of the prediction accuracy. The ego vehicle adjusted its speed to minimize the collision risk based on the predicted maneuvers. It was demonstrated that the AACC improved the driving safety compared with human drivers and the state-of-the-art previous ACC systems. As future work, we plan to develop a new filtering method for the stable estimation of operation characteristics. In addition, the relationship between the aggressiveness of a lane-changing mode and the operation characteristics in car-following should be investigated.

**Author Contributions:** Conceptualization, H.W., H.M., K.S., Y.T., A.Y., and H.A; methodology, H.W.; software, H.W.; validation, H.W.; formal analysis, H.W.; investigation, H.W.; resources, Y.T., H.M., and A.Y.; data curation, H.W.; writing—original draft preparation, H.W.; writing—review and editing, Y.T., H.M., and A.Y.; visualization, H.W.; supervision, Y.T., H.M., and A.Y.; project administration, K.S. and H.A.; funding acquisition, A.Y. and H.A.

**Funding:** This research received no external funding.

**Conflicts of Interest:** The authors declare no conflicts of interest.

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
