# Peer review of "Advanced Adaptive Cruise Control Based on Operation Characteristic Estimation and Trajectory Prediction†"

_applsci, doi:10.3390/app9224875_

Round 1

Reviewer 1 Report

This paper proposed an advanced ACC considering the collision risk. The authors applied GM model to predict the longitudinal movement of adjacent vehicles. The results show that the proposed method outperforms human drivers in terms of collision risk. There are a couple of major concerns.

The introduction does not discuss existing ACC models and point out the limitation of them.

There are several models are proposed:

C. Davis, “Effect of adaptive cruise control systems on traffic flow,” Phys. Rev. E - Stat. Nonlinear, Soft Matter Phys., vol. 69, no. 6 2, pp. 1–8, 2004.

Treiber, A. Hennecke, and D. Helbing, “Congested Traffic States in Empirical Observations and Microscopic Simulations,” Phys. Rev., vol. 62, no. 2, pp. 1805–1824, 2000.

Milanés and S. E. Shladover, “Modeling cooperative and autonomous adaptive cruise control dynamic responses using experimental data,” Transp. Res. Part C Emerg. Technol., vol. 48, pp. 285–300, 2014.

Lu and A. Aakre, “A new adaptive cruise control strategy and its stabilization effect on traffic flow,” Eur. Transp. Res. Rev., vol. 10, no. 2, 2018.

In section 3.2, authors are suggested to discuss the motivation of using the GM model to predict human driver behavior. Why?

The authors applied the Levenberg-Marquardt algorithm to get the value of three model parameters in GM model. What is the computational cost of this method? Delay is very crucial for autonomous driving strategies.

Numbers of lane change models have been proposed. Why do authors choose the sinusoidal model?

Li, X., & Sun, J. Q. (2017). Studies of vehicle lane-changing dynamics and its effect on traffic efficiency, safety and environmental impact. Physica A: Statistical Mechanics and its Applications467, 41-58.

For the performance evaluation, authors are suggested to compare the proposed method with both human driving behavior and the state-of-the-art autonomous driving strategies.

Author Response

We appreciate your helpful comments and suggestion regarding our work. We have revised the manuscript as recommended and are submitting a new version with tracked changes.

Reviewer 2 Report

This paper proposes an advanced driver assistance model that aims to minimize the risk of a collision due to lane changing behavior of aggressive/inattentive drivers. The model proposes adjustments to trajectory of a vehicle while minimize the risk of a rear-end collision and a side-swipe collision. The methodology is built on well-known transportation  models   such as GM model.  The proposed methodology is applied to real-world data collected on Germen highways and evaluated using F1 index to predict lane change maneuver  and RMSE to predict trajectory of vehicle.

This paper would be stronger if the authors discuss “Daganzo C.F., The cell transmission model: A dynamic representation of highway traffic consistent with the hydrodynamic theory, Transportation Research Part B: Methodological, Volume 28, Issue 4, August 1994, Pages 269-287” and “Wiedemann Car Following Model ” http://onlinepubs.trb.org/onlinepubs/conferences/2011/RSS/3/Higgs,B.pdf

What are the limitations of the proposed methodology? For example, how the model would operate if there are several autonomous vehicles following each other?  The model is proposed for SAE level 5 automation. Does the model operate in lower levels of automation?

Minor revision: please use “crash” instead of “accidents”. Accident implies that the event is random and not preventable. Roadway crashes are on the other hand preventable.

Line 200, 201, 202: define parameters of equations 1, 2, 3.

 Abstract should report performance measures, such as F1 score and RMSE obtained from the model.

Author Response

(The authors gave the same response as above.)

Round 2

Reviewer 1 Report

My concerns are addressed.